# Development and Differentiation of Epididymal Epithelial Cells in Korean Native Black Goat

**DOI:** 10.3390/ani10081273

**Published:** 2020-07-25

**Authors:** Yu-Da Jeong, Yun-Jae Park, Yeoung-Gyu Ko, Sung-Soo Lee, Sang-Hoon Lee, Jinwook Lee, Kwan-Woo Kim, Sung Woo Kim, Bongki Kim

**Affiliations:** 1Department of Animal Resources Science, Kongju National University, Yesan 32439, Korea; wjddbek202@naver.com (Y.-D.J.); gunin446@naver.com (Y.-J.P.); 2Animal Genetic Resource Research Center, National Institute of Animal Science, RDA, Hamyang 50000, Korea; kog4556@korea.kr (Y.-G.K.); lee6570@korea.kr (S.-S.L.); sanghoon@korea.kr (S.-H.L.); koreatop5@korea.kr (J.L.); bgring@korea.kr (K.-W.K.)

**Keywords:** basal cell, clear cell, epididymis, immunofluorescence

## Abstract

**Simple Summary:**

Studies have revealed that the communication networks among epididymal epithelial cells play an essential role in sperm maturation and storage. Therefore, the localization and expression of V-ATPase and cytokeratin 5 in the clear cells and basal cells, respectively, of immature and mature goat epididymis was investigated. To the best of our knowledge, the present study is the first to use immunofluorescence labeling and confocal imaging to characterize the differentiation of clear cells and basal cells in the goat epididymis. The findings could help the understanding of the regulatory mechanism that creates an optimal luminal environment for sperm maturation and storage in the epididymis.

**Abstract:**

The acidic luminal environment of the epididymis is regulated by the communication networks among epididymal epithelial cells; it is necessary for sperm maturation and storage. To characterize epididymal epithelial cell differentiation, the localization and expression of hydrogen-pumping vacuolar ATPase (V-ATPase) and cytokeratin 5 (KRT5) in the clear and basal cells, respectively, of immature and mature goat epididymis and vas deferens was examined. The epididymides and vas deferens were obtained from goats aged 1, 2, and 12–14 months. To assess the localization and expression patterns of V-ATPase and KRT5 in the caput, corpus, and cauda of the epididymis and proximal vas deferens, the tissue sections were subjected to immunofluorescence labeling and observed by confocal microscopy. Both clear and basal cells progressively started to differentiate in a retrograde manner. Clear cells disappeared from the cauda region after puberty, and they were maintained only in the caput and corpus regions of the adult goat epididymis. V-ATPase and KRT5 were co-expressed in the differentiated cells located at the base of the epithelium (i.e., basal cells). This cell type-specific differentiation and distribution of the epithelial cells plays a critical role in establishing a unique luminal environment for sperm maturation and storage in the goat epididymis.

## 1. Introduction

The epididymis is a highly convoluted duct that connects the testis and vas deferens (VD), and it is anatomically divided into the following three regions: the caput, corpus, and cauda. The epididymal epithelium consists of three major cell types, namely the principal cells (PCs), clear cells (CCs), and basal cells (BCs), which function cooperatively to establish an optimal luminal environment for the maturation, storage, protection, and transport of spermatozoa [1,2]. Principal cells are the most abundant cell type in the epididymis. They secrete various proteins for sperm maturation and protection, and reabsorb and secrete bicarbonate and water via the cystic fibrosis transmembrane conductance regulator (CFTR) [3,4] and Aquaporin 9 (AQP9) [5,6], respectively, in the apical membrane of the epithelium. Clear cells are located throughout the epididymis and secrete protons into the lumen via V-ATPase located in their apical membrane [7]. The activity of V-ATPase contributes to luminal acidification in several other organs, including the kidney, lung, and inner ear [7,8,9,10]. In the epididymis, its activity is critical for establishing a unique environment that maintains spermatozoa in a quiescent state during their maturation and storage. In addition, CCs appear to be involved in the disposal of contents of cytoplasmic droplets detached from spermatozoa [11]. Basal cells, which are located at the base of the epithelium, are dome-shaped, but in rodents, BCs extend long cytoplasmic cell body projections into the lumen of the epididymis [12,13]. Recent studies have reported that BCs can regulate proton secretion in neighboring CCs in a paracrine manner and control the luminal environment [12]. Overall, crosstalk and collaboration among the three types of epithelial cells in the epididymis are necessary for regulating and maintaining the environment for sperm maturation and storage [14]. Therefore, studies on the development and differentiation of epithelial cells will help understand the mechanisms of cell–cell communication involved in the establishment of luminal environment in the epididymis.

Recently, we reported that in pigs, the development and differentiation of the epididymal epithelium are determined by cell-specific mechanisms during postnatal development and that the establishment of an appropriate environment via the interaction of CCs and BCs is essential for sperm maturation and storage [15]. However, information regarding the development and differentiation of epididymal epithelium in goats is limited. Furthermore, identifying the mechanisms of development and differentiation mechanism of epithelial cells may be a critical step in understanding how the unique luminal environment in the epididymis is formed and regulated. Therefore, the purpose of this study was to investigate the expression and localization of V-ATPase and cytokeratin 5 (KRT5) to characterize the differentiation of CCs and BCs in the goat epididymis, respectively.

## 2. Materials and Methods

### 2.1. Tissue Collection

Epididymides ( the caput, corpus, and cauda) and VD tissues were collected from immature (1 month old, *n* = 5; 2 month old, *n* = 5) and mature (12–14 month old, *n* = 5) Korean native black male goats with average body weights of 4.07 ± 0.2, 6.51 ± 0.34, and 26.44 ± 2.62 kg, respectively, in the National Institute of Animal Science. There was no difference in samples, collected during the breeding season, between 12- and 14-month-old goats. Therefore, all epididymis samples collected from goats of these ages were considered as one group. All study procedures were reviewed and approved by the Institutional Animal Care and Use Committee at the National Institute of Animal Science (Approval No. NIAS 2018-290).

### 2.2. Tissue Fixation and Preparation

After harvest, the epididymides and VD samples were fixed by immersion in 4% paraformaldehyde dissolved in phosphate-buffered saline (PBS) for 24 h at room temperature, and then washed for 20 min with PBS for five times. Subsequently, the tissues were then incubated in a solution of 30% sucrose in PBS for at least 24 h, embedded in OCT compound (Tissue-Tek; Sakura Finetek, Torrance, CA, USA), mounted on a cutting block, and frozen. The tissue sections (10 µm thick) were obtained using a Leica 3050S cryostat (Leica Microsystems, Bannockburn, IL, USA), placed on the Fisher Superfrost/Plus microscope slides (Fisher Scientific, Pittsburgh, PA, USA), and stored at −20 °C until use.

### 2.3. Immunofluorescence Staining

The cryo-preserved tissue sections were hydrated in PBS for 10 min, and then microwave heated in 10 mM Tris/1 mM EDTA buffer (pH 9.0) for antigen retrieval (three 2-min treatments with a 5-min interval between treatments). Non-specific binding was blocked by incubation with 1% bovine serum albumin in PBS for 60 min at room temperature. The sections were then incubated with the primary antibodies in a moist chamber for 90 min at room temperature or overnight at 4 °C. Thereafter, the samples were washed in PBS and incubated with the secondary antibodies for 60 min at room temperature. Thereafter, the samples were washed and then placed in Vectashield medium (Vecta Labs, Burlingame, CA, USA) containing 4′, 6-diamidino-2-phenylindole (DAPI) to label the nuclei. As a negative control, the sections were incubated with non-immune serum (DAKO, Carpinteria, CA, USA) instead of the primary antibody. All antibodies used in the present study are listed in Appendix A (primary antibodies) and Appendix A (secondary antibodies) [1,13,16]. These were diluted in an antibody diluent (DAKO, Carpinteria, CA, USA). All confocal images were acquired using a LSM800 confocal microscope and analyzed using Zen Blue software (both from Zeiss, Oberkochen, Germany). For confocal 3D reconstruction, Z-series images were captured using a LSM800 confocal microscope. In addition, for adult tissues, mosaic images were obtained using the automated module of the LSM800 confocal microscope. The final images were generated using Adobe Photoshop software (Adobe Inc., San José, CA, USA).

### 2.4. Quantification of CCs and BCs

The number of CCs and BCs was determined as the number of cells that were positive for B1-VATPase and KRT5, respectively, per square millimeter of epithelial area. At least three epididymides were examined at each of the designated ages. Digital images were acquired using a ×20 objective with a Zeiss confocal microscope and were analyzed using Zen Blue software.

### 2.5. Statistical Analysis 

At each sampling time points, the epididymides from at least three individuals were examined, and for each epididymis, three cryostat sections were analyzed. A one-way analysis of variance was performed to identify significant differences between groups. A *p* value of <0.05 was used to indicate a significant difference. The epithelial area was measured using Zen Blue (lite) software and Image J software (http://www.imagej.nih.gov).

## 3. Results

### 3.1. Expression of B1-VAPTase and KRT5 in the Goat Epididymis

To confirm whether B1-VAPTase and KRT5 can be used as markers for CCs and BCs, respectively, the epididymal goat tissues were double labeled with these markers. While B1-VATase was exclusively expressed in the cytoplasm of CCs (Figure 1A, white arrows) and at the base of the epithelium (Figure 1A, yellow arrowheads), and KRT5 was expressed in the cytoplasm of BCs located at the base of the epithelium (Figure 1B, white arrowheads). In addition, BCs labeled with KRT5 were also labeled with B1-VATPase in the adult epididymis (Figure 1C). Labeling was not detected when the B1-VATPase and KRT5 primary antibodies were not included in tissue preparations (Figure 1D–F).

### 3.2. Changes in CCs of Immature and Mature Goat Epididymis

At postnatal month (PNM) 1, B1-VATPase-labeled CCs were observed throughout the epididymis and VD (Figure 2A–D), although they were rarely detected in the caput (Figure 2A). Within the VD, CCs were narrow and located in the inner epithelium (Figure 2D, arrows), whereas B1-VATPase-positive cells were located in the outer epithelium (Figure 2D, arrowheads). In the epididymis, CCs were round and goblet-like and occasionally detected in the caput and corpus/cauda, respectively (Figure 2A–C, arrows). However, B1-VATPase positive cells were not observed at the base of the epididymis epithelium (Figure 2A–C). At PNM2, in the epididymis, goblet-shaped CCs were detected in the corpus and cauda (Figure 2F,G, arrows), whereas round CCs were detected in the caput (Figure 2E, arrows). Clear cells were not observed in the inner epithelium of the VD (Figure 2H), but B1-VATPase-labeled cells were observed at the base of the epithelium of the cauda and VD (Figure 2G,H, arrowheads). At PNM14, several goblet-shaped CCs were observed in the caput and corpus (Figure 2I,J, arrows), but these cells were not observed in the cauda and VD (Figure 2K,L). In contrast, B1-VATPase-labeled BCs were observed at the base of the epithelium of all regions of the epididymis and VD (Figure 2I–L, arrowheads). Although CCs appeared progressively, in an ascending pattern from the distal region (cauda) to the proximal region (caput) of the epididymis (Figure 2), they were not detected in the cauda of the epididymis and VD. Clear cells tended to change their morphology from round- to goblet-shaped during differentiation, with age (Figure 2A,E,I). In addition, the changes in the numbers of CCs were examined according to region and age (Figure 2M). The quantification analysis showed that the number of CCs increased with age in the caput and corpus, but not in the cauda. These observations indicated that the differentiation of CCs is initiated in the distal region of the epididymis and proceeds to the proximal region.

### 3.3. Changes in BCs of Immature and Mature Goat Epididymis

At PNM1, BCs were observed in the VD and all regions of the epididymis (Figure 3A–D). Most BCs in the VD were triangular with short cytoplasmic projections and were located beneath other epithelial cells (Figure 3D, arrowheads). Basal cells with long narrow cytoplasmic body projections were also detected in the cauda of the epididymis (Figure 3C, arrowheads). Notably, the nuclei of BCs in the caput and corpus of the epididymis were located at the same height as those of the adjacent epithelial cells. Basal cells presented columnar and triangular morphologies in the caput and corpus, respectively (Figure 3A,B, arrowheads). At PNM2, most BCs in the VD and cauda of the epididymis were dome-shaped without cytoplasmic projections and were located beneath other epithelial cells (Figure 3G,H). Basal cells in the cauda of the epididymis appeared in the outer region of epithelium compared with those observed at PNM1 (Figure 3C,G). In the corpus region, most BCs had a long and narrow body projection that extended between adjacent epithelial cells toward the luminal border of the epithelium (Figure 3F). The nuclei of BCs tended to migrate to the base of the epithelium. The morphology and localization of BCs were similar to those observed in the cauda of the epididymis at PNM1 (Figure 3C,F), but BCs in the caput were columnar and at the same position as neighbor epithelial cells. At PNM14, most BCs from the epididymis and VD were dome shaped and were located only at the base of the epithelium (Figure 3I–L). However, some BCs were columnar resembling that of adjacent epithelial cells, some had the nucleus located in the basal pole of the epithelium and evidenced narrow body projections toward the lumen, and some were dome-shaped and located beneath other epithelial cells. To determine whether BC projections could pass through tight junctions and be in direct contact with the lumen, KRT5 was double labeled with ZO1, a tight junction protein (Figure 4) located in the most apical region of adhesion between epithelial cells (Figure 4, yellow arrowheads). Some BCs presented cytoplasmic extensions that reached the apical pole of the epithelium during the immature stages (PNM1 and PNM2) of the epididymis (Figure 4A–C). However, in the adult stages, BCs remained dome shaped and were located beneath adjacent epithelial cells (Figure 4D). The number of BCs peaked at PNM 2 and then significantly decreased at PNM 14 in all regions of the epididymis. Basal cells were maintained in the cauda as well as in the caput and corpus in PNM14, unlike CCs (Figure 3M).

### 3.4. Co-Localization of B1-VATPase and KRT5 in the Differentiated BCs

To determine whether B1-VATPase, known as a marker of CCs, is expressed in BCs, double-labeling was performed with B1-VATPase and KRT5. Co-localization was not observed in the entire epididymis at PNM1 (Figure 5A–C) and in the caput and corpus of the epididymis at PNM2 (Figure 5E,F). However, B1-VAPTase-positive BCs were detected in the VD at PNM1 (Figure 5D), in the cauda and VD at PNM2 (Figure 5G,H), and in the entire epididymis and VD at PNM14 (Figure 5I–L). Interestingly, B1-VATPase-positive BCs were detected only when these were located beneath other epithelial cells (differentiated). These results indicated that BCs in the epididymal epithelium exhibit a diverse range of morphologies and protein expression patterns that tended to be age and region dependent. A summary of co-localization of B1-VATPase and KRT5 in the epididymis and VD in immature (PNM1 and PNM2) and mature (PNM14) goats is shown in Table 1.

## 4. Discussion

In the present study, immunohistochemistry and antibodies directed against B1-VATPase and KRT5 were used to investigate the localization and morphology of CCs and BCs in the epididymis of immature and mature goats. To the best of our knowledge, this is the first study to describe the development and differentiation of the epididymal epithelial cells in goats using cell-specific markers.

### 4.1. Differentiation of CCs in the Goat Epididymis

Clear cells were columnar, narrow, and goblet shaped (Figure 6) during epithelial development, but only goblet-shaped CCs were present in the caput and corpus of the epididymis after differentiation. These results are consistent with those of a previous study showing that CCs show different morphologies in the developing epididymis, but only one shape in the differentiated epididymis of pigs [15]. In contrast, rodent CCs show different shapes, such as pencil-, cuboidal-, and goblet-shaped, across the epididymal regions [7,17]. It has been proposed that the morphological development of CCs is species specific. In this regard, testicular luminal factors (TLFs) might be candidate regulators of the morphological changes observed in the goat CCs. TLFs synthesized and secreted from the testis make direct contact with epididymal epithelial cells [13,18]. However, according to the present results, the morphological changes of CCs appeared early in the developing epididymis. A previous study indicated that the secretion of testicular luminal fluid may occur only in the normal mature testis of many species [19]. Therefore, TLFs might not be the regulators of the morphological changes in CCs. In addition, CCs were observed in all regions of the epididymis during development, but maintained only in the caput and corpus of the epididymis after puberty. In contrast, previous studies indicated that CCs are absent in the goat epididymis [20]. The discrepancy between the present and previous study findings could be explained by the different procedures used to detect epididymal epithelial cells. While in the present study, cell-specific markers were used to identify individual cell types, other studies have classified cell types based on their morphological characteristics by hematoxylin and eosin staining. The identification of morphological differences between epithelial cells in the goat epididymis is difficult because of the dynamic changes in their morphologies with age and regions (Figure 2 and Figure 3). Thus, further studies are needed to improve our understanding of the detailed regulatory mechanisms involved in the changes in the localization and morphology of CCs. 

### 4.2. Differentiation of BCs in the Goat Epididymis

As illustrated in Figure 6, the development and differentiation of BCs seems to occur first in the VD, and then in the cauda and finally in the caput of the epididymis; thus, they are initiated in a retrograde manner. In addition, BCs were observed in all regions of the epididymis at PNM1, although their number was very low in the caput and corpus regions of the epididymis. This finding is not in consistent with that of previous studies, which reported that BCs were only present in the epididymis as early as PNM2 although they were found in the VD [21]. As described for CC differentiation, it is difficult to distinguish the types of epithelial cells based only on their morphology. Thus, the hematoxylin and eosin staining procedure is limited to observing the development and differentiation of epididymal epithelial cells, particularly in the early stages when there are no localization differences among the multiple cell types, including PCs, CCs, and BCs (Figure 6). Using a KRT5 antibody that specifically labels BCs in a range of species, including mice [13], rats [22], bats [23], and pigs [15], we were able to distinguish different types of epididymal epithelial cells. Furthermore, dynamic morphological changes in BCs were observed with age and across regions of the epididymis. Undifferentiated BCs were columnar and located at the same height as those of the adjacent epithelial cells (Figure 3 and Figure 5). Thereafter, BCs moved close to the basal lamina of the epithelium and were narrower or triangular. Finally, after differentiation, BCs were hemispherical and their nuclei were located at the base of the epithelium (Figure 3 and Figure 5). These observations are in agreement with those of previous studies on the pig epididymis [15]. The shapes of BCs in the epididymis varies greatly among species. In rodents, the projected BCs are maintained after completion of their differentiation and play an important role in scanning the luminal environment [12,24], whereas in other species, such as bats [23] and pigs [15], they disappear. These results suggest that BCs’ regulation and function may be species-specific. An important finding of the present study was that B1-VATPase, known as a specific marker for CCs, was expressed in both differentiated BCs and CCs of the goat epididymis. Recent studies have described that the interaction among epithelial cells is necessary for luminal acidification, and V-ATPase is known to play an important role in luminal acidification, which is crucial for sperm maturation and the subsequent storage in a quiescent state [25,26]. To the best of our knowledge, this is the first study to propose that BCs may directly contribute to luminal acidification through V-ATPase. However, further research is needed to fully understand the role of V-ATPase in the epididymal BCs of goat.

## 5. Conclusions

The present study results, revealed that both CCs and BCs progressively start to differentiation in a retrograde manner, starting at the VD and then progressing into the cauda, followed by the more proximal segments of the goat epididymis. During development and differentiation, CCs disappear from the cauda and exclusively remain in the caput and corpus of the epididymis. Although B1-VATPase was known to be expressed only in CCs; here, it was expressed in the differentiated BCs of the goat epididymis. These results indicate that the appropriate differentiation and localization of epididymal epithelial cells are necessary to control and regulate luminal acidification in the goat epididymis. Further studies are warranted to gain a better understanding of the mechanisms underlying the regulation of luminal acidification by communicating between CCs and BCs in the goat epididymis. 

## Figures and Tables

**Figure 1 animals-10-01273-f001:**
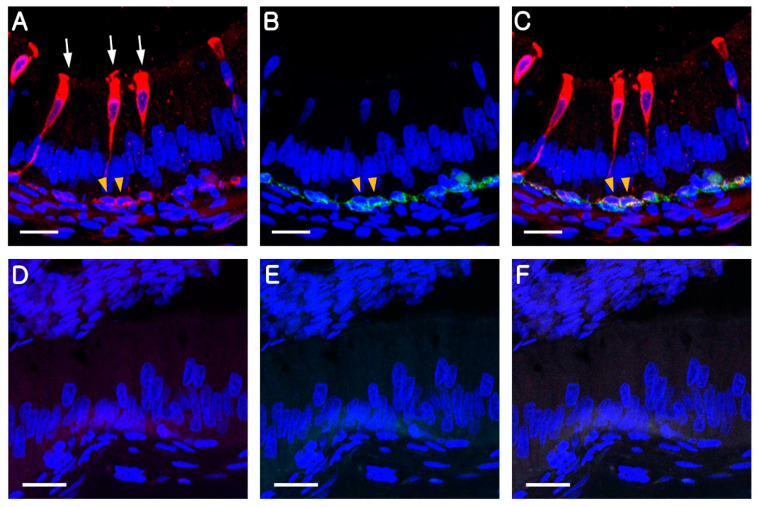
Immunolocalization of B1-VATPase (red) and KRT5 (green) in the epididymis of goat at postnatal month (PNM) 14. (**A**) Epididymis section labeled with anti-B1-VATPase. Goblet-shaped clear cells (CCs) were observed (arrows). (**B**) Epididymis section labeled with anti-KRT5. BCs were located at the base of the epithelium (arrowheads). (**C**) BCs were double labeled for B1-VATPase and KRT5 (arrowheads). (**D–F**) Negative controls showed no B1-VATPase or KRT5 staining in the epididymis. The nuclei were labeled with DAPI (blue). Bars = 20 µm.

**Figure 2 animals-10-01273-f002:**
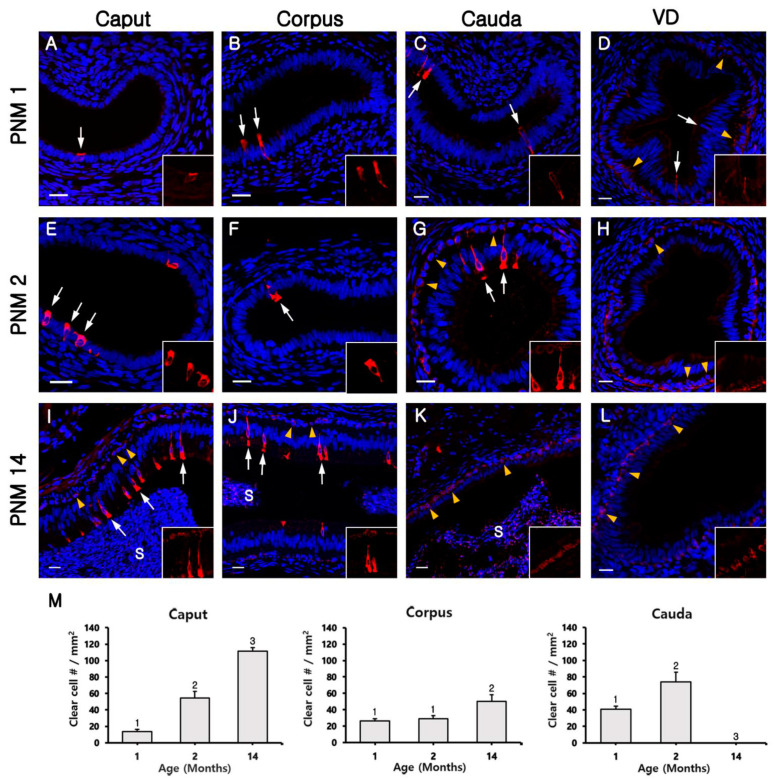
Progressive appearance of CCs from the vas deferens (VD) to the caput of the goat epididymis. Clear cells in the epididymis were labeled for B1-VATPase (red). (**A**–**D**) The epididymis at postnatal month (PNM) 1. (**E**–**H**) The epididymis at PNM2. (**I**–**L**) The epididymis at PNM14. The arrows and arrowheads indicate positive epithelial cells in the inner and outer epithelium, respectively. S, spermatozoa. The nuclei were labeled with DAPI (blue). Bars = 20 µm. (**M**) Changes in the proportion of CCs in the caput, corpus, and cauda. The cell were calculated based on the number of CCs in the epithelium per square millimeter of the epididymis area. The results are expressed as the mean ± standard error of the mean. Different letters indicate significant differences (*p* < 0.05).

**Figure 3 animals-10-01273-f003:**
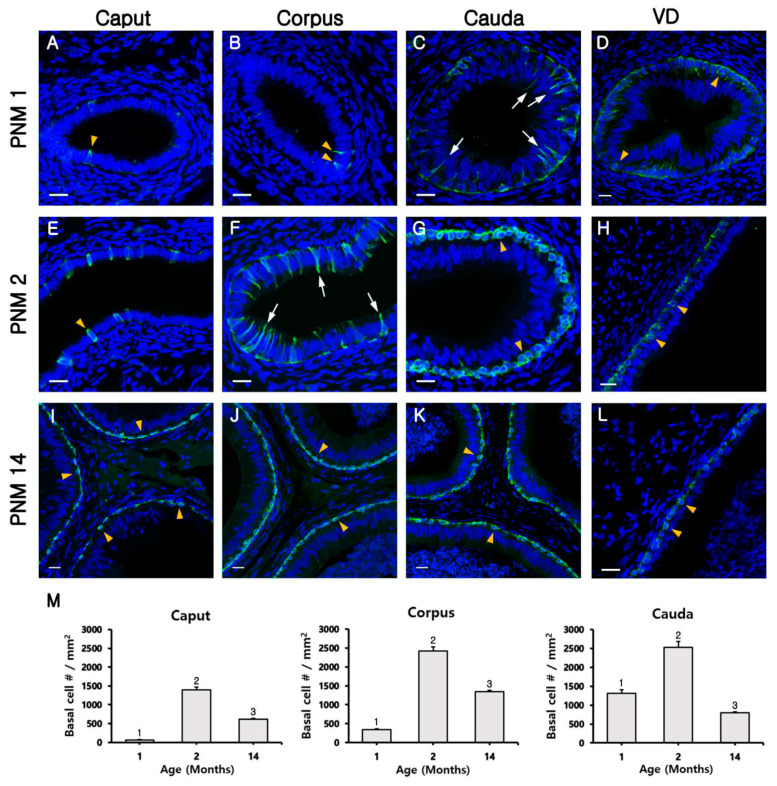
Progressive appearance of basal cells (BCs) from the vas deferens (VD) to the caput of the goat epididymis. Basal cells in the epididymis were labeled for KRT5 (green). (**A**–**D**) The epididymis at postnatal month (PNM) 1. (**E**–**H**) The epididymis at PNM2. (**I**–**L**) The epididymis at PNM14. The arrows and arrowheads indicate projecting and non-projecting BCs, respectively. The nuclei were labeled with DAPI (blue); S, spermatozoa. Bars = 20 µm. (**M**) Change in the volume of BCs in the caput, corpus, and cauda. Cell numbers were calculated based on the number of BCs in the epithelium per square millimeter of the epididymis area. The results are expressed as the mean ± standard error of the mean. Different letters indicate significant differences (*p* <0.05).

**Figure 4 animals-10-01273-f004:**
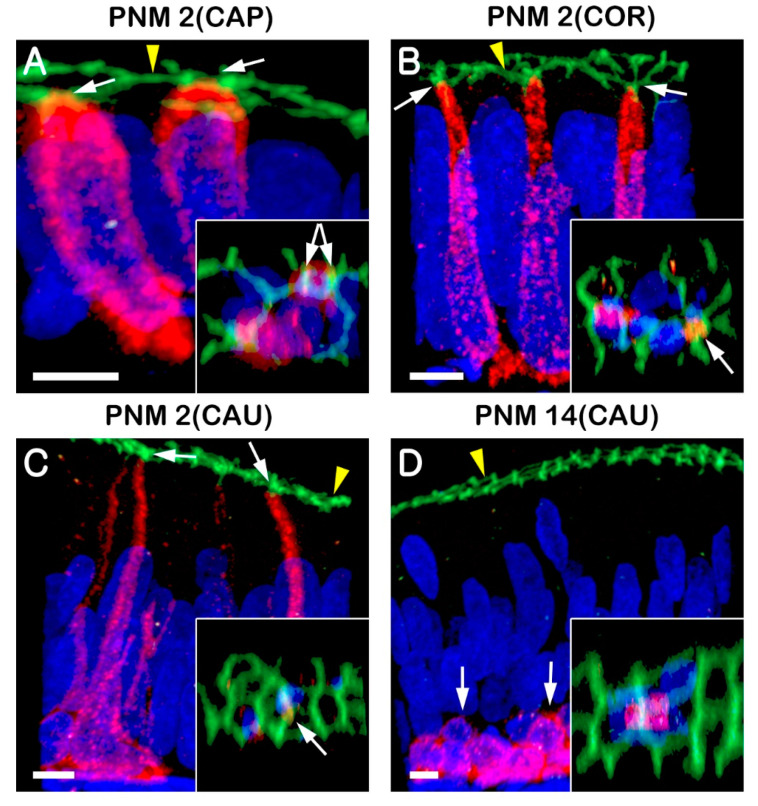
Basal cell projections in the goat epididymis. Basal cells were labeled for KRT5 (red), and tight junctions for ZO1 (green). During the first 2 months of postnatal development: (**A**) columnar-shaped BCs in contact with the tight junction labeled by ZO1 (arrows). (**B**,**C**) Basal cell projections passed through the tight junctions coming into contact with the luminal contents (arrows). (**D**) The arrows indicate BCs with no projections. The arrowheads indicate tight junctions. The insets show projected BCs to reach the tight junction (arrows). The nuclei were labeled with DAPI (blue). Bars = 5 µm.

**Figure 5 animals-10-01273-f005:**
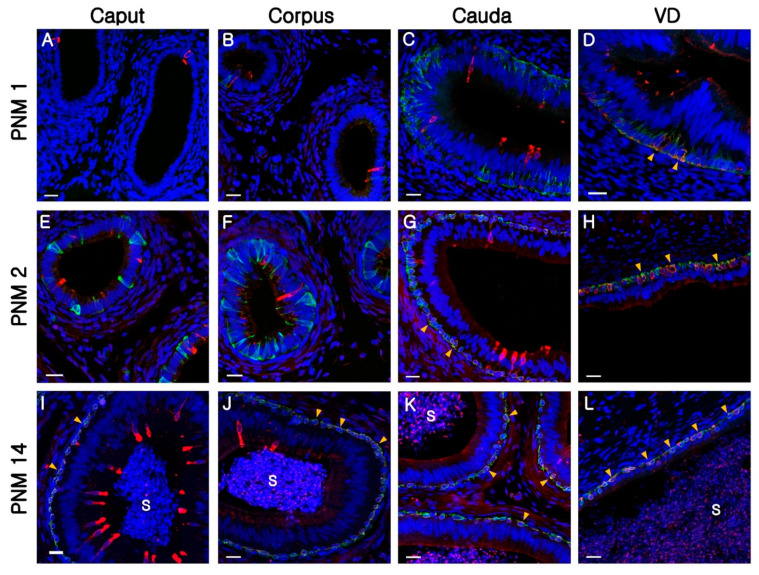
Co-localization of B1-VATPase and KRT5 in the goat epididymis and VD. Cells in the epididymis were double labeled for B1-VATPase (red) and KRT5 (green). (**A**–**D**) The epididymis at postnatal month (PNM) 1. (**E**–**H**) The epididymis at PNM2. (**I**–**L**) The epididymis at PNM14. The nuclei were labeled with DAPI (blue). The arrowheads indicate BCs double labeled for B1-VATPase and KRT5. Bars = 20 µm.

**Figure 6 animals-10-01273-f006:**
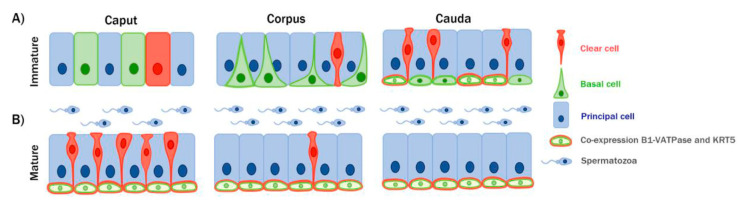
Schematic diagram of the epididymal epithelium differentiation in immature and mature goats. (**A**) During the immature stage, in the caput region, all epithelial cells have a similar morphology (columnar shaped). In the corpus region, CCs are goblet shaped and their nuclei begin to migrate to the apical area of the epithelium. Basal cells have narrow- or triangular- shaped cytoplasmic cell bodies and their nuclei begin to migrate to the base of the epithelium. In the cauda region, all CCs are goblet shaped and their nuclei are located in the apical area of the epithelium. All BCs are dome-shaped and located at the base of the epithelium. B1-VATPase is expressed in some BCs. (**B**) During the mature stage, goblet CCs were found in the caput and corpus regions of the epididymis but not in the cauda region. Basal cells located at the base of the epithelium co-expressed KRT5 and B1-VATPase in all regions of the epididymis.

**Table 1 animals-10-01273-t001:** Co-localization of B1-VATPase and KRT5 in the epididymis and VD of immature and mature goat.

Postnatal Month (PNM)	Co-Localization of B1-VATPase and KRT5			
	CA	CO	CD	VD
PNM1	−	−	−	+
PNM2	−	−	+	+
PNM14	+	+	+	+

CA, caput; CO, corpus; CD, cauda; VD, vas deferens; +, positive; −, negative.

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
