# Peer review of "Development and Differentiation of Epididymal Epithelial Cells in Korean Native Black Goat"

_animals, 2020, doi:10.3390/ani10081273_

Round 1

Reviewer 1 Report

17th June 2020

Development and differentiation of epididymal 2 epithelial cells in Korean native black goat

In the present work, the aim was to assess the localization and expression patterns of V-ATPase and KRT5 in the different the caput, corpus, and cauda of the epididymis, in order to understand the development and differentiation of epididymal epithelial cells in black goats.

General comments

The manuscript was well written, the references are actual and the approach was very complete. The subject is opportune and original.

However, extensive English revision and correction is required.

Specific comments

Abstract

-The abstract needs to be improved as in the present form, there is no information about the number of epididymis that were analysed. Besides, the abstract finalises without a clear conclusion, only with descriptions. Authors found interesting differences on the clear and basal along prepubertal puberty. So, some explanation even summarised, should be added.

Introduction

Introduction is complete and focused in the cells of epididymis.

Material and methods

-In material and methods, the same information missing in the abstract is missing here

-L41 to L43: extensive English revision is required.

Results

In results section, besides qualitative information, could a grade system be adopted in order to better quantify and characterize the expression of the specific markers?

Figures have adequate quality and are very well identified.

Discussion

-Consider to place Figure 6, although interesting in the comprehension of the study, in another section such as Material and Methods or Results, not in Discussion section.

-L241 to 244: English revision is required in order to improve phrase construction. For instance, "non-motile" sperm could be better attributed than immotile.

-Figure 6: the small size of the draws (clear cell, basal cell …) are evident, being difficult to discriminate them.

-L271: please improve English “undifferentiated”

Conclusions

This section is adequate, but the physiological implications of main findings of the present study are missing.

Author Response

The authors would like to thank the editor and reviewers for their helpful and thoughtful comments. We incorporated all the suggestions and comments into the manuscript.

Reviewer 2 Report

L.22 If you describe the abbreviation for V-ATPase, you do not need to explain it again. (L.45)

L.30 Homogenize in the text the term of B1-VATPase when describing the results, it refers to V-ATPase interchangeably.

L41: Do three major cell types PCs, CCs and BCs, secrete proteins for sperm maturation and protection and reabsorb and secrete bicarbonate?

Can the development of these types of cells (PCs, CCs and BCs) be mediated by hormonal factors? specifically in mature animals

L66: delete “..”

L70: Add in the manuscript "male" before "goats".

Include in the materials and methods section the sample number and the place where the experiment was carried out. How many animals were used for the research? I mean, is the number used representative for this type of study?

How was it determined that the animals were fully mature? just for age?

It is mentioned that the collection of the epididymis and RV tissue was carried out in immature (1 and 2 month-old) and mature (12 to 14 month-old) native Korean goats. However, in the results, only evidence from 1, 2 and 14-month-old males is shown, omitting the results of tissue sampling from 12 and 13-month-old male goats. Add the results of the samples of the missing males or correct the data in the materials and methods section.

In Figures 2 and 3, as well as in the text of the results, the term “postnatal month (PNM)” is used instead of mature and immature, homogenize the terms for clarity throughout the manuscript.

Specify in the materials and methods section, specifically in point 2.1 Tissue Collection, that tissue extraction was performed from all regions of the epididymis (caput, corpus and cauda) so that it is in accordance with the figures in the results.

Results section:

L178. In figure 3, in epididymis at postnatal month (PNM) 1 and Epididymis at PNM2. B-F (corpus) there is a lot of variability for KRT5 (green) in this sense, one month will be necessary to increase the number of BCS cells or is it due to the individual factor?

Discussion:

L218. You mention that “It has been proposed that the morphological development of CCs is species-specific. In this regard, testicular luminal factors (TLFs) might be candidate regulators of the morphological changes observed in the goat CCs.” but in line 228, you said “Therefore, TLFs might not be the regulators of the morphological changes of CCs.”

Mention data on other species ej. Pigs

At what latitudes did the study take place? because bucks are seasonal, can this influence the cells (PCs, CCs and BCs) of the epididymis or the conditions of sperm maturation?

Author Response

(The authors gave the same response as above.)

Reviewer 3 Report

The presented study showed the sites of localization of B1-VATPase and cytokeratin 5 (KRT5) in different regions of the epididymis and vas deferens during the postnatal development of gonads in the black Korean goat. The authors have used B1-VATPase and KRT5 as markers for differentiating the clear cells and basal cells in the epididymis respectively.

Although the manuscript is written well and the study design was comprehensive but I recommend that the manuscript can not be considered for the publication unless the author is willing to revise it.

Major concerns:

  1. A similar study has been reported previously in pigs (https://www.ajas.info/upload/pdf/ajas-19-0587.pdf) where they have mentioned the quantitative changes in the clear and basal cells in the epididymis during postnatal development. In the current study, authors have shown a few representative images from different regions of epididymis stained with B1-VATPase and KRT5 antibodies. I think it would be better if they can provide quantitative measurements about the types and number of cells in different regions during postnatal development.
  2. Authors also have not mentioned about the number of animal samples were used in the study and the number of experiments done to reach a significant rational based conclusion.
  3. I think that the results can be improved further in terms of sections-integrity for example Fig. 2K and L both look almost the same. It is difficult to say that which one is cauda and which one is vas deferens.
  4. DAPI is too strong in almost all the figures, making it difficult to judge the staining of B1-VATPase in the basal cells in the cauda and vas deferens (Fig. 2). Similarly In Fig. 2D, H, and I, the staining of the basal cells and the red-background all look the same, giving the staining as false positive. I think this can be improved either by reducing the background or increasing the staining intensity further.
  5. Base upon histology, vas deferens can be broadly divided into two different regions proximal and distal regions. In the current study, the authors show the sections from different regions in different stages of postnatal development can not be considered for the comparison. Fig. 2D and H look cross-section while Fig. 2L seems to be a longitudinal section.
  6. Line 140, 141, and 142: The authors make a statement " Furthermore, in PNM14, B1-VATPase was highly expressed in the spermatozoa of the cauda compared to those in the caput of the epididymis (Fig. 2I and K)". I think it is not correct unless you isolate the sperm from the caput and cauda regions separately and stain them with B1-VATPase and see the difference. Because I suspect that it is the real staining since Fig. 2I has very high DAPI fluorescence compared to Fig. 2K.
  7. The size of scale bars in most of the figures is also not uniform.

Minor revisions:

  1. In the abstract, line 32: "caput to the cauda, B1-VATPase increases in the spermatozoa" this point should be removed unless it is not experimentally proved as mentioned above in major concerns.
  2. Line 42: Full name of CFTR is missing: Cystic fibrosis transmembrane conductance regulator
  3. Line 42: Authors should cite the following papers about one of the strong studies on the CFTR and AQP9 localization in the epididymis and vas deferens https://link.springer.com/article/10.1007/s10735-019-09813-3 and https://onlinelibrary.wiley.com/doi/full/10.1002/mrd.23317.
  4. Line: I think there is a technical error in several places. The luminal acidification of the epididymal tubule is required for the sperm maturation and storage of the sperm in a quiescent state. the authors mentioned only sperm maturation and storage.
  5. I feel presenting the antibodies and fluorescent tags in a tabular form is always much appreciated and clear.
  6. Many people use adobe photoshop software for preparing figures for papers, but considering the bit depth and gamma values, I personally don't recommend it. It's always better to use ZEN black, FIJI, or other scientific imaging software than these programs whose algorithms will change the scientific information from your figures.
  7. At several places, B1-VATPase is mentioned as B1-VAPase. Please correct that.
  8. Line 150, 151. The sentence should be re-written. It doesn't make any sense as of now.

Author Response

(The authors gave the same response as above.)

Round 2

Reviewer 1 Report

9th July 2020

Development and differentiation of epididymal 2 epithelial cells in Korean native black goat

In the present revision, authors have accomplished several improvements in scientific content and also in the writing style making the article much more readable.

General comments

The present revision answered to the main questions raised in the first version. Globally the abstract is improved as the material and methods. Some doubts in figure captions are however present in the revised version.

Specific comments

Material and methods has now reference to the number of epididymis that were used at each age group, which is beneficial to the reader.

L76-77: “We did not observe any different results among 12- to 14-month-old samples collected during breeding season”

This sentence should be improved, in order to avoid “we”. You could say that “no difference was found among (...)”

The figures captions have suffered some changes as they are now more complete, however it is difficult to follow the captions as there is reference to a (M) that is not seen as well as the comments to a statistical analysis that is not visible, for instance.

L176-180:

“(M) Change in the volume of basal cells in the caput, corpus, and cauda. Cell numbers were calculated based on the number of basal cells in the epithelium per square millimeter of the epididymis area. Results are expressed as the mean ± standard error of the mean. Different letters indicate significant differences (p <0.05)”.

L219-222:

“M) Change in the volume of CCs in the caput, corpus, and cauda. Cell numbers were calculated based on the number of CCs in the epithelium per square millimeter of the epididymis area. Results are expressed as the mean ± standard error of the mean. Different letters indicate significant differences (p <0.05). “

If these captions are corrected or the insertion clarified, and the previous comments fulfilled, I think the Editor may proceed to acceptance.

Author Response

We thank you for reviewing our manuscript.

Reviewer 2 Report

The research work has sufficient information in the summary and introduction sections. The section on materials and methods was improved, the author made the suggested changes and justify his answers. In addition, supplementary information was included. The wording of the discussion section was reorganized, is clear and explains the results. 

Author Response

(The authors gave the same response as above.)

Reviewer 3 Report

The manuscript is only corrected in terms of the text but there is no improvement in the figures and the data addition as claimed by the authors. I could not find the attached revised version of the new figures and there is no statistical data added anywhere.

Corrections:

1. Line 26, 27, and 28. To assess the localization and expression patterns of V-ATPase and KRT5 in the different the caput, corpus, and cauda of the epididymis, immunofluorescence labeling was performed and tissue sections were observed using confocal microscopy.

Please mention about the proximal vas deferens in the sentence, if it is taken from the caudal attached region.

2. Line 73: Space should be there before parentheses. If there are several double spaces, please correct that also.

3. If information about antibodies manufacturers and antibody dilutions were presented in the tabular form then why the author hasn’t removed it from the materials and methods section?

4. The author mentioned that they have revised the figure 2 and 3 based upon previous comments but I am sorry to say that there is no difference between the previous version of the manuscript and the revised version of the manuscript in regards to the figures 2 and 3. And at the same time, I do not see any statistical data that has been added to the manuscript.

5. The same goes for the figure 6 as well. I am not able to find the attached updated version of the figure.

6. V-ATPase again has spelled incorrectly at several places. For example Line: 130

7. The author is trying to suggest that clear cell expression goes down in a retrograde manner from Cauda to Caput during post during the post-natal development of the gonads. Based on the results shown in figure 2, their results do not support their hypothesis substantially. I strongly recommend providing the cell counts and the better images where they have a higher number of cells. To be very precise, at PNM1 the Corpus region has a higher number of cells which decrease at PNM2 but increase again the PNM14. This whole thing goes against their own hypothesis of the paper. Similarly, figure 2D showing the staining in the inner epithelium is not at all convincing.

8. The manuscript is poorly written in terms of the scientific quotient. At several places, it is not clear at all what the author is trying to convey.

9. I personally feel that the manuscript needs a significant revision in terms of the clarity of sentences. I strongly recommend using the professional service of manuscript editing if the authors are not capable of conveying their message clearly, concisely, and specifically.

Author Response

(The authors gave the same response as above.)

Round 3

Reviewer 3 Report

I appreciate authors efforts to make the changes in manuscripts. I don't require further revision of manuscript. Hence I will recommend to accept the manuscript in its present form.